# Time-dependent predictors of loss to follow-up in HIV care in low-resource settings: A competing risks approach

Tamrat Endebu Gebre◉*, Girma Taye, Wakgari Deressa

Department of Epidemiology and Biostatistics, School of Public Health, College of Health Sciences, Addis Ababa University, Addis Ababa, Ethiopia

* seetame2008@gmail.com

## Abstract

Loss to follow-up (LTFU) remains a major challenge in HIV care, particularly in resource-limited settings. While several studies have identified its predictors, many have neglected the competing risks of transfer out and death, as well as the dynamic influence of these predictors over time. A retrospective cohort study was conducted among adult HIV patients who initiated antiretroviral therapy (ART) between 2019 and 2024. LTFU was a primary outcome, whereas transfer out and death were competing risks. A Fine–Gray subdistribution hazard ratio (SHR) regression analysis identified LTFU predictors within a competing risk framework. An extended SHR model with a time–covariate interaction term was used to examine the predictors' time–varying effects on LTFU risk. Data analysis was performed via STATA 17 and Python 3.9. In a cohort of 4,135 HIV patients (8,521.54 person-years of follow-up), the overall incidence of LTFU was 13.10 per 100 person-years (95% CI: 12.35–13.89), with cumulative risks of 15%, 25%, and 35% at 1, 3, and 5 years post-ART, respectively. The predictors of LTFU included younger age (15–24 years: aSHR = 1.51), male sex (aSHR = 1.24), incomplete address details (aSHR = 1.72), noninitiation/noncompletion of TPT (aSHR = 2.16), poor adherence (aSHR = 2.54), and undernutrition (aSHR = 2.03). While younger age (e.g., 15–24 years) was associated with an increased risk of LTFU at baseline (baseline aSHR = 1.36, p = 0.014), this association diminished over time (interaction aSHR = 0.54, p = 0.001). Undernutrition consistently predicted LTFU (baseline aSHR = 1.64, p < 0.001), with no significant time-dependent effect (interaction aSHR = 1.01, p = 0.903). In conclusion, this study highlights the high incidence of LTFU among HIV patients and its key predictors. Notably, age has a significant time-dependent effect, with its influence on the risk of LTFU being most pronounced during the early stage of ART initiation, whereas nutritional status remains a consistent predictor of LTFU over time.

**Data availability statement:** All relevant data are within the manuscript and its Supporting Information files.

**Funding:** The author(s) received no specific funding for this work.

**Competing interests:** The authors have declared that no competing interests exist.

## Introduction

HIV (human immunodeficiency virus) care and treatment programs aim to achieve sustained viral suppression and improve patient survival through lifelong adherence to antiretroviral therapy (ART) [1]. However, patients lost to follow-up (LTFU) in HIV care remain a persistent challenge [2,3]. LTFU is defined as patients disengaging from their HIV care or ART appointment for more than 28 days from their appointment date [4]. The consequences of LTFU are multifaceted, affecting not only individual health outcomes but also the broader success of ART programs [5]. High rates of LTFU can lead to treatment failure, increased morbidity, mortality, and drug resistance, particularly in resource-limited settings [6], ultimately undermining public health efforts to control the HIV epidemic [7].

Previous studies estimated the rate of LTFU across different regions of Africa, such as 25.1% in South Africa [8], 23.9% in Nigeria [9], 39.8% in Congo [10], 26.7% in Uganda [11], 26% in Malawi [12], and 15.2% in Ethiopia [3]; overall, it ranges from 6.7% to 58.3% in sub-Saharan Africa [13]. Studies have also identified several predictors of LTFU, including age and sex [2,14], address information [11], adherence to ART, tuberculosis preventive therapy (TPT) status, WHO (World Health Organization) clinical stages, low CD4 counts [15–17], socioeconomic factors such as poverty and lack of social support, lack of nutrition, and psychosocial factors such as stigma and discrimination [18–20]. However, these studies have largely relied on traditional survival analyses, which may not adequately account for competing risks such as death and transfer out [21].

In HIV care, death and transfer out are competing events often observed as follow-up outcomes [21]. For example, a systematic review and meta-analysis conducted in Ethiopia reported significant rates of transfer out (11.17%) and death (6.75%) [3]. Since HIV-positive individuals who die or transfer out are no longer at risk of LTFU, failing to consider this competing risk analysis can result in overestimated LTFU rates and misinterpretation of the effects of predictors [21,22]. Notably, the relative importance of those predictors across different follow-up periods remains unclear, limiting the design of stage-specific interventions in HIV care. Some predictors may exert their strongest influence during early ART initiation, whereas others may become more relevant over time [23]. Given these gaps, we aimed to estimate LTFU incidence while accounting for competing risks, identify key predictors, and examine their time-dependent effects on LTFU risk in HIV care.

## Methods

### Study design and setting

A retrospective open cohort study was conducted in low-resource urban settings in central Ethiopia, including Addis Ababa and surrounding towns, where the HIV prevalence is estimated at 2.8–3.48% [24]. The study focused on high-volume ART hospitals, including Zewditu Hospital (7,644 patients currently on ART), ALERT Hospital (7,460), and Yekatit 12 Hospital (3,312) in Addis Ababa. Additionally, facilities from the surrounding towns in Oromia Regional State included Adama Teaching Hospital

(7,470), Bishoftu Hospital (4,005), Geda Health Center (1,395), Adama Health Center (1,837), and Asella Hospital (2,200). All eligible patients from these facilities were included in the study.

## Participants

The study included patients aged 15 years or older who initiated ART between July 1, 2019, and April 30, 2024, with at least one follow-up visit. As an open cohort, patients were enrolled on a rolling basis as they became eligible throughout the study period. Patients were excluded if they had transferred in (TI) with incomplete baseline data, had unavailable medical charts during data collection, or had unknown ART initiation dates. A census sampling approach was used to include all eligible patients.

## Variables

*The primary outcome variable* was loss to follow-up (LTFU), defined as failure to attend the next scheduled HIV clinic appointment or medication refill visit within ≥28 consecutive days of the scheduled date, without documentation of transfer out or death, or medical discontinuation of ART [25]. This definition reflects national HIV program guidance and accommodates variations in visit schedules, including multimonth dispensing (MMD) models, where patients may not return for routine visits between scheduled refills. For patients with clearly documented refill intervals, the reference point for LTFU classification was the expected refill date. In cases where refill dates were not explicitly recorded, the standard 1–3 month follow-up schedule was assumed on the basis of the local ART regimen and clinical guidelines [25].

*The secondary outcome (competing risk) variables,* documented transfer out (relocation to another health facility with formal documentation) and recorded death, were treated as competing events [25].

*Time Variable*: Time-to-event was measured in person-years of observation (PYO) for each patient. This interval spans from ART initiation to the earliest occurrence of LTFU, transfer out, death, or administrative censoring (end of follow-up: April 2024). Patients without events were right-censored at their last recorded clinic visit [26].

*Predictor variables:* This study examined sociodemographic and clinical predictors of LTFU. Sociodemographic factors included *sex* (male vs. female), *age* at enrollment, and *address information*. Address information was categorized as *green* if the patient's contact details were complete and current, including a phone number and a specific kebele address with a house number, or y*ellow* if the information was incomplete or outdated. Patients' clinical factors were recorded at the time of last contact for retained patients, whereas for those who were lost to follow-up, died, or transferred out, data were recorded at the time of last contact or enrollment. Clinical factors included *adherence*, with good adherence defined as taking ≥95% of doses (≤2 missed per month for 30-day regimens; ≤ 3 missed for 60-day regimens), and poor adherence, defined as taking <85% of doses (>5 missed per month for 30-day regimens; > 9 missed for 60-day regimens). Additionally, the *WHO Clinical Stage* was categorized as Stage I/II for early/moderate disease and Stage III/IV for advanced disease (adolescents/adults with CD4 < 200 cells/mm$^3$). *TB preventive therapy (TPT) status* was classified as *gold* for completed TPT, *silver* for initiated but not completed, and *bronze* for not started. Finally, *nutritional status* was categorized as underweight, normal, or overweight on the basis of weight-for-height/age criteria [27].

## Data collection and quality assurance

A multicenter electronic medical record dataset comprising all eligible participants who initiated ART between July 1, 2019, and April 30, 2024, was accessed for research purposes between May 1 and May 15, 2024. Data extraction followed a structured tool aligned with Ethiopia's National HIV Care/ART Forms [28] and was pretested at a nonstudy ART facility to ensure reliability. Trained data collectors extracted deidentified records under the supervision of two research supervisors. Two-day training was conducted to ensure consistency in data abstraction, management, and confidentiality. To minimize bias, the data collectors and staff were blinded to the outcome variables. Data quality was monitored via SmartCare-ART's built-in assurance features, which addressed duplicates, missing values, and inconsistencies [29].

## Data processing and analysis

The data were initially recorded in the spreadsheet and then exported to statistical software for further cleaning, coding, and analysis. Exploratory analysis addressed missing values. Descriptive statistics summarizing the baseline cohort characteristics. The incidence rates for LTFU and competing events (transfer out, death) were calculated as the number of events per 100 person-years at risk. Person time represented the sum of individual follow-up durations until a terminating event (LTFU, transfer out, death) or censoring (study end). Competing risk analysis treats each event type as a distinct outcome [26].

*Cumulative incidence function (CIF) analysis:* The CIF estimates the probability of an event occurring over time when competing risks exist—events that preclude the event of interest [30]. Group differences in CIFs were assessed via Gray's test, a nonparametric method designed for competing risks [31].

*Competing risk regression analysis*: In accordance with established methodological guidelines [32], we employed a dual analytical strategy combining cause-specific hazard ratio (CHR) regression and Fine and Gray subdistribution hazard ratio (SHR) regression models [33,34]. The CHR model assesses the effect of covariates on the risk of LTFU by treating competing events as censored observations [33]. Conversely, the Fine and Gray SHR model directly accounts for competing risks, quantifying the effect of covariates on the CIF of LTFU [34]. This dual approach facilitated a comprehensive understanding of predictor effects within the competing risk framework. Adjusted SHR values along with their 95% confidence intervals (CIs) were reported for each predictor.

*Time-dependent analysis:* To examine how covariates and the risk of LTFU change over time, we added time-covariate interaction terms to the Fine and Gray SHR model [35]. To validate the results, we complemented the analysis with a sensitivity analysis using Fine and Gray models stratified by the follow-up period. Analyses were conducted via STATA 17 (for competing risk regression via *stcrreg* and CIF estimation with *stcompet* [36] and Python 3.9 (for lifelines and competing risk libraries) [37].

## Ethical issues

This study was approved by the ethical review board (IRB) of the College of Health Sciences (CHS) at Addis Ababa University (AAU), under reference number 061/23/SPH. All data were fully anonymized prior to access, and the requirement for informed consent was waived by the ethics committee, as the study involved retrospective analysis of deidentified data. We strictly maintained confidentiality by using anonymized records solely for research purposes.

## Results

### Characteristics of a cohort

The cohort consisted of 4,135 HIV patients with a median age of 39 years (IQR: 31–46). Approximately one-third of the patients were aged 25–34 years (28.6%), 35–44 years (33.6%), or 45 years or older (29.7%). The majority were female (59.6%), and most had complete address information, including a phone number (categorized as "green") (84.5%). In terms of TB prophylaxis treatment (TPT), 62.6% had completed treatment (gold status), whereas 37.4% had either not started treatment or not completed treatment (bronze/silver status). With respect to differentiated service delivery (DSD) models, 30.0% were enrolled in the appointment spacing (ASM)/6-month multimonth dispensing (6MMD) model, 24.0% were included in the 3MMD model, and 46.0% were either not enrolled or enrolled in other forms. Adherence to ART was good in 64.3% of patients, whereas 35.7% had poor adherence. Nutritional status varied, with 41.9% having normal nutrition, 39.1% being undernourished, and 19.0% being overweight. The cohort was nearly evenly split into WHO clinical stages I/II (53.0%) and III/IV (47.0%). Over a median follow-up period of 1.67 years (IQR: 0.67–3.42), 51.7% of patients were censored (alive), 27.0% were LTFU, 12.9% transferred out, and 8.4% died Table 1.

**Table 1. Cohort characteristics of HIV-positive adults in Ethiopia (2019–2024) (n = 4135).**

| Variables | Frequency (n) | Percentage (%) | Median (IQR) |
|---|---|---|---|
| **Age in continuous** | | | 39 (31, 46) |
| **Age in category** | | | |
| 15-24 | 335 | 8.1 | |
| 25-34 | 1183 | 28.6 | |
| 35-44 | 1390 | 33.6 | |
| 45+ | 1227 | 29.7 | |
| **Sex** | | | |
| Male | 1672 | 40.4 | |
| Female | 2463 | 59.6 | |
| **Address information** | | | |
| Green | 3496 | 84.5 | |
| Yellow | 639 | 15.5 | |
| **TPT status** | | | |
| Gold | 2587 | 62.6 | |
| Bronze/silver | 1548 | 37.4 | |
| **DSD Model** | | | |
| ASM/6MMD | 1239 | 30.0 | |
| 3MMD | 994 | 24.0 | |
| Not enrolled/Other forms* | 1902 | 46.0 | |
| **Adherence** | | | |
| Good | 2657 | 64.3 | |
| Poor | 1478 | 35.7 | |
| **Nutritional status** | | | |
| Normal | 1733 | 41.9 | |
| Undernourished | 1617 | 39.1 | |
| Overweight | 785 | 19.0 | |
| **WHO clinical stage** | | | |
| Stage I/II | 2191 | 53.0 | |
| Stage III/IV | 1944 | 47.0 | |
| **Follow-up time in years** | | | 1.67 (0.67, 3.42) |
| **Outcome status** | | | |
| Censored (alive) | 2136 | 51.7 | |
| LTFU | 1116 | 27.0 | |
| Transfer out | 534 | 12.9 | |
| Death | 349 | 8.4 | |

Abbreviations: ASM = Appointment spacing model; DSD = Differentiated service delivery. IQR = Interquartile range, LTFU = Loss to follow-up, MMD = Multi-month dispensing, TPT = TB prevention therapy, WHO = World Health Organization, Others** = other DSD models such as community-based models, key populations, adolescents, and young people, and PMTCT (Prevention of Mother-to-Child Transmission) models.

### Incidence rates for LTFU and competing risks

A total of 4,135 HIV patients who commenced ART within the past five years were observed, contributing a cumulative total of 8,521.54 person-years until LTFU or transfer out or death. Among these patients, 1,116 were LTFU, resulting in an incidence rate of 13.10 per 100 person-years (95% CI: 12.35, 13.89); 534 patients were transferred out to other health facilities with an incidence rate of 6.27 per 100 person-years (95% CI: 5.76, 6.82); and 349 patients died, yielding an incidence rate of 4.10 per 100 person-years (95% CI: 3.69, 4.55) Table 2.

**Table 2. Incidence rates for LTFU and competing risks among HIV-positive adults, Ethiopia (2019–2024).**

| Events | Number | Total person years at risk | Incidence rates per 100 PYO (95%CI) |
|---|---|---|---|
| LTFU | 1116 | 8,521.54 | 13.10 (12.35, 13.89) |
| Transfer out | 534 | 8,521.54 | 6.27 (5.76, 6.82) |
| Death | 349 | 8,521.54 | 4.10 (3.69, 4.55) |

Abbreviations: CI = confidence interval, LTFU = loss to follow-up, PYO = person-year observation.

## Cumulative incidence of LTFU and competing risks

The cumulative incidence plots revealed that the probabilities of LTFU risk after ART initiation were approximately 15% at one year, 25% at three years, and 35% at five years; the probabilities of transfer out after ART initiation were approximately 7% at one year, 15% at three years, and 25% at five years; and the probabilities of death were estimated at 5% at one year, 8% at three years, and 10% at five years Fig 1.

## Predictors of LTFU risk

First, univariate analyses were performed via both cause-specific hazard regression (CHR) and Fine and Gray subdistribution hazard regression (SHR) models to examine the crude associations between each candidate covariate and the risk of LTFU S1 Table in S1 File. The multivariable CHR and SHR models were subsequently fitted, adjusting for all the candidate variables. The adjusted models identified age, sex, address, TPT status, adherence, nutritional status, and WHO clinical stage as significant predictors of LTFU (p < 0.05). Compared with those aged 45–24, 25–34, and 35–44 years, HIV patients aged 15–24, 25–34, and 35–44 years had an increased risk of LTFU by 51%, 80%, and 28%, respectively (aSHR: 1.51; 95% CI: 1.19, 1.91), (aSHR: 1.80; 95% CI: 1.53, 2.11), and (aSHR: 1.28; 95% CI: 1.09, 1.50). Sex was also significantly associated with LTFU, with male patients showing a 24% greater risk than female patients (aSHR: 1.24; 95% CI: 1.09, 1.41). Patients with incomplete address details (e.g., missing phone numbers or house addresses) had a 72% increased risk of LTFU (aSHR: 1.72; 95% CI: 1.49, 2.01). Patients who had not initiated or completed TB preventive therapy (TPT) were at more than twice the risk of LTFU as those who had completed TPT (aSHR: 2.16; 95% CI: 1.89, 2.46). Patients with poor adherence and those classified as undernourished presented significantly greater risks of LTFU, with SHRs of 2.54 and 2.03, respectively (aSHR: 2.54; 95% CI: 2.23, 2.89) and (aSHR: 2.03; 95% CI: 1.76, 2.32). Patients classified as WHO Stage III/IV had a 55% lower risk of LTFU than those classified as Stage I/II (aSHR: 0.45; 95% CI: 0.39, 0.51) Table 3.

## Time-dependent effects of predictors

Table 4 presents the Fine and Gray SHR model results with time-varying interactions. The main effects show baseline associations with LTFU risk, whereas interactions assess changes over time. Significant interactions (p < 0.05) indicate time-dependent effects, whereas nonsignificant interactions (p ≥ 0.05) suggest stable SHRs [35,38]. Younger age groups had higher baseline SHRs for LTFU than did those aged 45+ (15–24 years: aSHR = 1.36, p = 0.014; 25–34 years: aSHR = 1.63, p < 0.001; 35–44 years: aSHR = 1.20, p = 0.024). However, interaction terms revealed that this effect diminished over time (15–24 years: interaction aSHR = 0.54, p = 0.001; 25–34 years: interaction aSHR = 0.75, p = 0.023; 35–44 years: interaction aSHR = 0.70, p = 0.003). Patients with noninitiated or uncompleted TPT had a greater baseline risk (baseline aSHR = 1.24, p = 0.001), but this effect decreased over time (interaction aSHR = 0.63, p = <0.001). Poor adherence strongly increased the baseline risk (baseline aSHR = 2.03, p < 0.001), but the protective effect of good adherence decreased over time (interaction aSHR = 0.73, p < 0.001). Male gender (baseline aSHR = 1.15, p = 0.028; interaction aSHR = 1.08, p = 0.200), incomplete address information (baseline aSHR = 1.62, p < 0.001; interaction aSHR = 1.01,

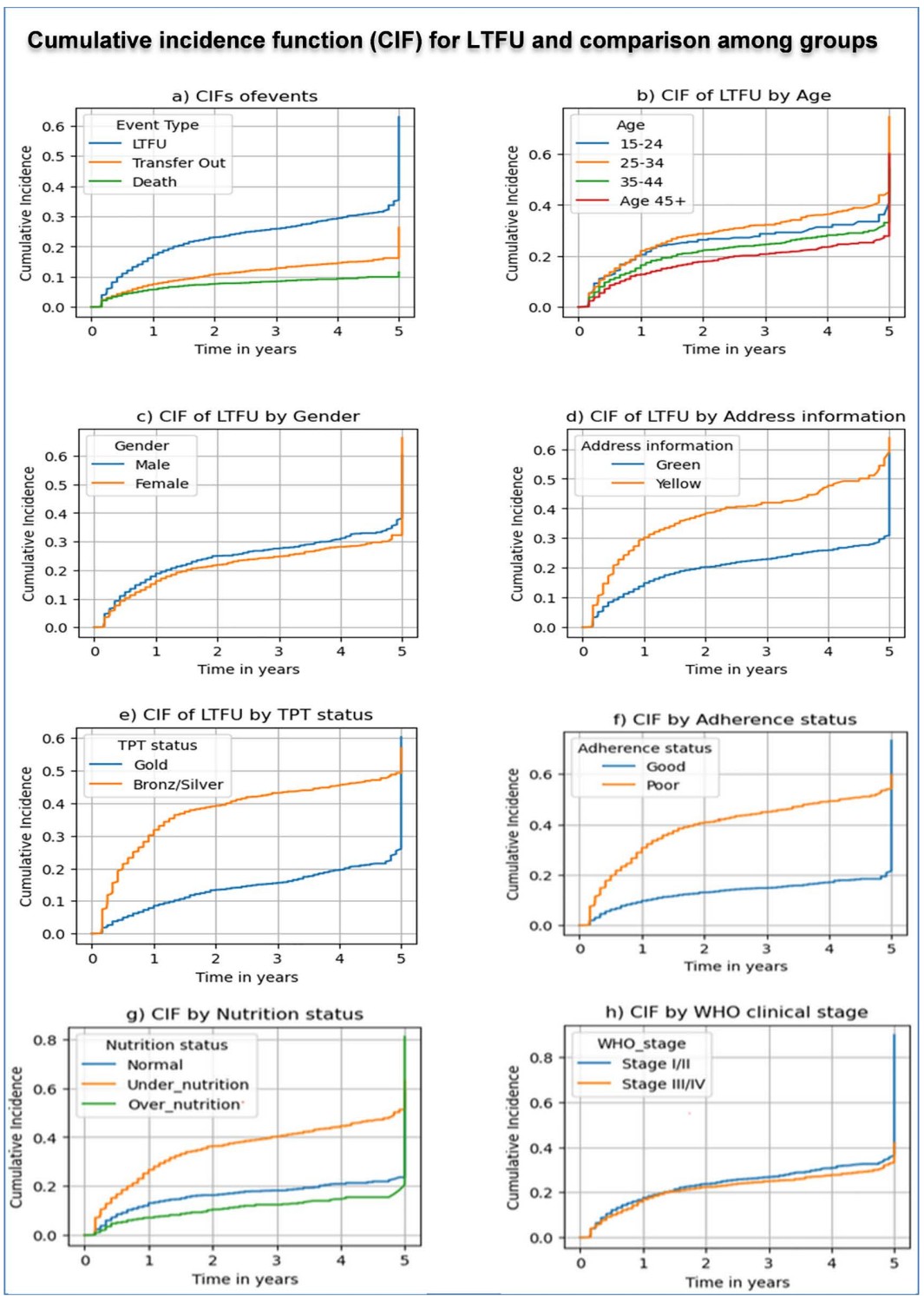

**Fig 1. Cumulative incidence functions (CIFs) of LTFU and comparisons across different groups.** Abbreviations: CIF = cumulative incidence function, LTFU = loss to follow-up, TPT = tuberculosis prevention therapy, WHO = World Health Organization.

**Table 3. Cause-specific and subdistribution hazard regression analysis to identify predictors of LTFU among HIV-positive adults in Ethiopia (2019–2024).**

| Covariates | LTFU (N = 1116) | TO (N = 534) | Death (N = 349) | Competing risk regression models | |
|---|---|---|---|---|---|
| | | | | aCHR (95% CI) | aSHR (95% CI) |
| **Age category** | | | | | |
| 15-24 | 99 | 45 | 16 | 1.45 (1.14, 1.83)** | 1.51 (1.19, 1.91)** |
| 25-34 | 399 | 189 | 53 | 1.84 (1.56, 2.17)*** | 1.80 (1.53, 2.11)*** |
| 35-44 | 353 | 167 | 130 | 1.25(1.06, 1.47)** | 1.28 (1.09, 1.50)** |
| 45+ | 265 | 133 | 150 | Ref | Ref |
| **Sex** | | | | | |
| Male | 490 | 198 | 191 | 1.25 (1.11, 1.42)** | 1.24 (1.09, 1.41)*** |
| Female | 626 | 336 | 158 | Ref | Ref |
| **Address information** | | | | | |
| Green | 838 | 448 | 306 | Ref | Ref |
| Yellow | 278 | 86 | 43 | 1.64 (1.43, 1.89)*** | 1.72 (1.49, 2.01)*** |
| **TPT status** | | | | | |
| Gold | 452 | 247 | 83 | Ref | Ref |
| Bronze/silver/ | 664 | 287 | 266 | 2.99 (2.62, 3.41)*** | 2.16 (1.89, 2.46)*** |
| **Adherence** | | | | | |
| Good | 412 | 225 | 163 | Ref | Ref |
| Poor | 704 | 309 | 186 | 2.87 (2.51, 3.30)*** | 2.54 (2.23, 2.89)*** |
| **Nutrition status** | | | | | |
| Normal | 328 | 204 | 128 | Ref | Ref |
| Undernourished | 683 | 270 | 172 | 1.87 (1.61, 2.16)*** | 2.03 (1.76, 2.32)*** |
| Overweight | 105 | 60 | 49 | 0.77 (0.62, 0.96)* | 0.79 (0.64, 1.01) |
| **WHO clinical stage** | | | | | |
| I/II | 612 | 110 | 65 | Ref | Ref |
| III/IV | 504 | 424 | 284 | 0.54 (0.48, 0.61)*** | 0.45 (0.39, 0.51)*** |

Abbreviations: aCHR = Adjusted Cause-specific Hazard Ratio, aSHR = Adjusted Subdistribution Hazard Ratio, CI = Confidence interval, LTFU = Loss to Follow-Up, Ref = Reference, TPT = TB Prevention Therapy, TO = Transfer Out, WHO = World Health Organization. = P value < 0.05, * = P value < 0.01, *** = P value < 0.001.

p = 0.891). Undernutrition was consistently associated with an increased risk of LTFU (baseline aSHR = 1.64, p < 0.001; interaction aSHR = 1.01, p = 0.903) Table 4. Time-stratified sensitivity analysis confirmed that all younger age groups had the strongest association with LTFU risk only in the first year after ART initiation (15–24 years: aSHR = 1.77, 95% CI: 1.35–2.32; 25–34 years: aSHR = 1.52, 95% CI: 1.24–1.86; 35–44 years: aSHR = 1.36, 95% CI: 1.11–1.65). Undernutrition remained a strong predictor of LTFU across all time intervals (0–12 months: aSHR = 1.57, 95% CI: 1.33–1.84; 13–36 months: aSHR = 2.71, 95% CI: 2.06–3.56; 37–60 months: aSHR = 2.46, 95% CI: 1.57–3.86) S2 Table in S1 File.

## Discussion

This study revealed a substantial incidence of LTFU among HIV patients on ART, with the risk progressively increasing over time. Although the specific incidence rates and cumulative probabilities are detailed in the results section, our findings are broadly comparable to those reported in similar settings, including Uganda (12.7% PYO) [39], Northwest Ethiopia (13.45% PYO) [40], and Southern Ethiopia (11.9% PYO) [41]. Similarly, the cumulative probabilities of LTFU at one and five years were consistent with those reported in prior studies [42,43]. However, the overall LTFU rate in this study was

**Table 4. Fine and Gray SHR model with time-covariate interaction terms to assess time-varying effects on LTFU risk among HIV-positive adults, Ethiopia (2019–2024).**

| Covariates | aSHR (95% CI) | P Value | Remarks |
|---|---|---|---|
| **Age category (years)** | | | Main effects |
| 15-24 | 1.36 (1.06, 1.74) | 0.014 | |
| 25-34 | 1.63 (1.38, 1.91) | <0.001 | |
| 35-44 | 1.20 (1.02, 1.42) | 0.024 | |
| 45+ | Ref | | |
| **Sex** | | | |
| Male | 1.15 (1.01, 1.30) | 0.028 | |
| Female | Ref | | |
| **Address information** | | | |
| Green | Ref | | |
| Yellow | 1.62 (1.42, 1.86) | <0.001 | |
| **TPT status** | | | |
| Gold | Ref | | |
| Bronze/silver/ | 1.24 (1.08, 1.41) | 0.001 | |
| **Adherence** | | | |
| Good | Ref | | |
| Poor | 2.03 (1.78, 2.31) | <0.001 | |
| **Nutritional status** | | | |
| Normal | Ref | | |
| Undernourished | 1.64 (1.43, 1.88) | <0.001 | |
| Overweight | 0.93 (0.74, 1.18) | 0.604 | |
| **Age*time** | | | Interaction term effects |
| 15-24 | 0.54 (0.39, 0.77) | 0.001 | |
| 25-34 | 0.75(0.59, 0.96) | 0.023 | |
| 35-44 | 0.70 (0.54, 0.88) | 0.003 | |
| **Gender*time** | | | |
| Male | 1.08 (0.95, 1.22) | 0.200 | |
| **Address information*time** | | | |
| Green | 1.01 (0.87, 1.16) | 0.891 | |
| **TPT status*time** | | | |
| Gold | 0.63 (0.55, 0.72) | <0.001 | |
| **Adherence*time** | | | |
| Good | 0.73 (0.64, 0.83) | <0.001 | |
| **Nutritional status*time** | | | |
| Normal | 0.96 (0.78,1.19) | 0.761 | |
| Undernureshed | 1.01 (0.83, 1.23) | 0.903 | |

Remarks: Main effects = covariate effects at baseline (time = 0) on cumulative incidence functions (CIFs) of LTFU risk; interaction term effects = how the effect of a covariate changes over time [35,38].

lower than those reported in Uganda (26.7% PYO) [11], Congo (33.48% PMO) [10], and Malawi (26% PYO) [12]. Conversely, the LTFU incidence rate in this study was higher than that in previous reports from northern Ethiopia (6.7% PYO) [44] and southern Ethiopia (5.3% PYO) [45]. The observed variations in LTFU rates across settings may be attributed to several factors, including differences in healthcare system efficiency, socioeconomic instability, stigma, ART adherence

challenges [46], and variations in the definition of LTFU across studies [47]. Furthermore, the incorporation of competing risk analysis may influence LTFU estimates by providing a more accurate risk assessment than standard survival analysis methods do [48].

### Predictors of LTFU and their time-dependent effects

This analysis revealed that younger age groups were at significantly greater risk of LTFU than older adults were. This finding was consistent with the findings of regional studies across sub-Saharan Africa [17,18,49,50], which revealed that younger individuals presented particularly elevated LTFU risks during transitions to older age groups [51]. Moreover, time-dependent and time-stratified analyses demonstrated that younger age among HIV-positive patients is significantly associated with LTFU risk in a time-dependent manner, with the effects most pronounced during the first year of ART initiation. This temporal pattern aligns with broader epidemiological trends observed in sub-Saharan Africa, where early ART discontinuation is disproportionately high among younger individuals [52]. LTFU incidence peaks within the first 6–12 months of ART, with younger patients (<35 years) facing the highest risk during this period [53,54]. This critical period coincides with challenges in treatment adjustment, including medication side effects and psychosocial stressors [55]. Thus, age-specific support during the first year, leveraging peer networks or mobile health reminders, and psychosocial counseling could mitigate LTFU among young people [51].

This study revealed a gender disparity in loss to follow-up, with male patients experiencing a slightly greater risk than females do, which is consistent with other regional studies reported from Uganda [11], sub-Saharan Africa [14], Tanzania [56], South Africa [17], Nigeria [15], and West Ethiopia [57]. Men's tendency to disengage from HIV care is often attributed to traditional gender roles, limited health-seeking behaviors, and perceived stigma [58], which can lead to reduced engagement with healthcare services. Additionally, employment-related migrations and barriers to regular appointments exacerbate this issue [18,59]. Time-dependent analyses revealed no significant changes in the effect of sex on LTFU risk over time, suggesting that these disparities persist throughout the treatment course. Hence, to address these challenges, implementing gender-sensitive retention strategies, such as enhanced communication and follow-up mechanisms (e.g., phone reminders, community health worker visits), and flexible service delivery models can help mitigate these disparities and improve overall retention in HIV care programs [60,61].

This study revealed a significant association between incomplete address details such as phone numbers or house addresses and a greater risk of loss to follow-up. This finding was consistent with findings from previous studies in Uganda and other regions, where the absence of telephone contact was linked to increased LTFU [49,62]. The interaction term between address information and time was not significant, suggesting that the effect of address completeness on LTFU risk remains stable over time. However, a time-stratified sensitivity analysis revealed that the effect was most pronounced during the initial treatment phase. A possible explanation might be that health facilities often rely on accurate contact information to trace patients and remind them of appointments, underscoring the importance of maintaining up-to-date address details [63]. Therefore, providing address information and periodic verification can help mitigate the risk of losing contact with patients over time, emphasizing the critical role of accurate contact details in enhancing retention within HIV care programs [64].

This analysis revealed a significantly elevated risk of loss to follow-up among HIV-positive individuals who did not initiate or complete TB preventive therapy, which is consistent with findings from previous studies in North Ethiopia [43], southern Ethiopia [41], Northeast Ethiopia [43], and Gondar [42]. The protective effect of TPT completion aligns with meta-analytic evidence demonstrating a 64% reduction in TB incidence and 26% lower mortality among people living with HIV (PLHIV) who complete preventive therapy [65]. The time-dependent analyses revealed a declining protective effect of TPT completion over the follow-up duration. This is possibly due to missed opportunities for TB screening and preventive therapy integration at earlier stages of care [66]. Therefore, ensuring early TPT initiation and completion and early follow-up systems can improve retention and strengthen overall outcomes in HIV care programs.

This study identified poor adherence to ART as a strong predictor of LFTU, which aligns with previous findings from Tanzania [56], Gondar [44], southern Ethiopia [41], and Northwest Ethiopia [40]. Poor adherence often correlates with missed clinic visits, limiting opportunities for follow-up and engagement with healthcare providers [67,68]. Time-dependent analyses revealed that the protective effect of good adherence diminishes over time, yet sensitivity analyses revealed that poor adherence remained a robust predictor of LTFU across all follow-up periods. This could be possible, as early LTFU dynamics, including challenges in treatment adaptation and psychosocial stressors, contribute to elevated LTFU risks during initial treatment phases [50]. Strengthening adherence support mechanisms from early stages and maintaining them over time can significantly improve retention and optimize treatment outcomes for patients on ART [69].

This study identified undernutrition as a strong predictor of LTFU risk among adults living with HIV. These findings align with evidence from Malawi [12], Uganda [62], Tanzania [50], and multiple Ethiopian cohorts [3,20,70]. In this study, time-dependent analyses confirmed that nutrition was a stable predictor of LTFU risk across follow-up periods. Undernutrition is linked to accelerated disease progression, increased opportunistic infections (OIs), and a 32.5% reduction in OI incidence with nutritional interventions [71]. The bidirectional relationship between HIV and undernutrition, where immune compromise exacerbates malnutrition and poor nutrition increases mortality and tuberculosis risk, creates a cycle of deteriorating health and care disengagement [72]. Food insecurity and poverty further compound these challenges, limiting consistent clinical access [20,70]. We strengthened the recommendation that integrated nutritional strategies, such as therapeutic feeding programs and routine malnutrition screening, are needed throughout the ART treatment course [19].

Finally, in this study, we identified a protective effect of advanced WHO clinical stages (III/IV) against LTFU, with patients in these stages demonstrating a significantly lower risk of loss than those in earlier stages (I/II). This finding aligns with few previous studies reported from Zimbabwe (aHR = 0.67) [73] and Ethiopia (WHO clinical stage III (HR = 0.6) and IV (HR = 0.8)) [54]. However, it contrasts with several other studies, such as those from Uganda (aHR = 1.50) [49], Tanzania (aHR = 1.8) [56], and Ethiopia (OR = 1.85) [3]. The observed protection may stem from intensified clinical monitoring for advanced-stage patients, including frequent follow-ups (e.g., quarterly vs. biannual visits), adherence counseling, and nutritional support [74]. Conversely, patients in earlier stages may disengage due to perceived wellness, underestimating the need for consistent care [75]. Further research is warranted to explore this finding.

## Limitations

First, we acknowledge the potential for misclassification due to undocumented outcomes, such as unrecorded deaths or silent transfers, which may have been incorrectly classified as LTFU. This limitation could introduce bias in estimating the true incidence of LTFU and may affect the accuracy of associations identified between predictors and LTFU. Second, we recognize inherent methodological constraints that may affect the precision of estimating both the magnitude and time-varying effects of predictors on LTFU. To address these issues, we applied a dual analytical strategy—combining the CHR and SHR regression models—to provide a comprehensive understanding of predictor effects. Furthermore, time-stratified sensitivity analyses were conducted to address potential violations of proportionality assumptions in the Fine and Gray model [38]. This combined approach within a competing risks framework enhances the robustness and interpretability of our findings despite these methodological limitations.

## Conclusions

This study revealed a high incidence of LTFU among HIV patients, with the cumulative incidence indicating a rising probability of LTFU over time after accounting for transfers out and death. Younger age, male sex, incomplete address information, noninitiated or uncompleted TB preventive therapy (TPT), poor adherence, and undernutrition were significant predictors of LTFU risk. Importantly, the age of HIV patients was significantly time dependent, indicating that its effects on LTFU risk were pronounced only during the first year of ART initiation. Nutritional status remains a stable predictor of LTFU risk over time, showing no significant temporal variation.

## Implications and recommendations

To reduce LTFU and enhance HIV care outcomes, time-sensitive and targeted interventions are essential. Strategies should focus on high-risk subgroups, such as younger patients, those with poor adherence, and those with undernutrition. For predictors with time-varying effects, interventions should be tailored to specific follow-up periods. Conversely, stable predictors require sustained clinical attention throughout the treatment course. These recommendations align with the WHO's call for differentiated service delivery models while emphasizing the understudied role of time-aware clinical decision-making in low-resource HIV care settings [76].

## Supporting information

**S1 File. S1 Table.** Univariate CHR and SHR analyses to identify predictors of LTFU among HIV-positive adults in Ethiopia (2019–2024). Legends: Abbreviations: CHR = cause-specific hazard ratio, SHR = subdistribution hazard ratio, CI = confidence interval, LTFU = loss to follow-up, Ref = reference, TPT = TB prevention therapy, TO = transfer out, WHO = World Health Organization. **S2 Table.** Fine and Gray SHR models stratified by follow-up period to assess time-varying effects on LTFU risk among HIV-positive adults, Ethiopia (2019–2024). Legends: Abbreviations: aSHR = Adjusted Sub-distribution Hazard Ratio, CI = Confidence interval, LTFU = Loss to Follow-Up, Ref = Reference, TPT = TB prevention Therapy; TO = Transfer Out, WHO = World Health Organization, * = P-value < 0.05, ** = P-value < 0.01, *** = P-value < 0.001. **S3 Data.** dta: De-identified dataset used for analysis.
(ZIP)

## Author contributions

**Conceptualization:** Tamrat Endebu Gebre, Wakgari Deressa.

**Data curation:** Tamrat Endebu Gebre.

**Formal analysis:** Tamrat Endebu Gebre.

**Funding acquisition:** Girma Taye.

**Investigation:** Tamrat Endebu Gebre, Girma Taye, Wakgari Deressa.

**Methodology:** Tamrat Endebu Gebre, Girma Taye, Wakgari Deressa.

**Supervision:** Girma Taye, Wakgari Deressa.

**Validation:** Girma Taye, Wakgari Deressa.

**Visualization:** Tamrat Endebu Gebre.

**Writing – original draft:** Tamrat Endebu Gebre.

**Writing – review & editing:** Tamrat Endebu Gebre, Girma Taye, Wakgari Deressa.

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
