## [Decision Letter · Decision Letter 0]

Dear Dr. Gebre,

Thank you for submitting your manuscript to PLOS ONE. After careful consideration, we feel that it has merit but does not fully meet PLOS ONE’s publication criteria as it currently stands. Therefore, we invite you to submit a revised version of the manuscript that addresses the points raised during the review process.

We look forward to receiving your revised manuscript.

Kind regards,

Hamufare Mugauri, Ph.D. Medicine and Health Sciences

Academic Editor

PLOS ONE

3. We note that your Data Availability Statement is currently as follows: [All relevant data are within the manuscript and its supporting information files.]

Reviewers' comments:

Reviewer's Responses to Questions

**Comments to the Author**

1. Is the manuscript technically sound, and do the data support the conclusions?

Reviewer #1: Yes

Reviewer #2: Yes

2. Has the statistical analysis been performed appropriately and rigorously?

Reviewer #1: Yes

Reviewer #2: Yes

3. Have the authors made all data underlying the findings in their manuscript fully available?

Reviewer #1: Yes

Reviewer #2: Yes

4. Is the manuscript presented in an intelligible fashion and written in standard English?

Reviewer #1: Yes

Reviewer #2: Yes

Reviewer #1: I have enjoyed reading your well written manuscript on time-dependent predictors of LTFU in Ethiopia. Please see a few comments for your consideration.

Data processing and analysis

1. Line 139 - Consider using plural for data. Data were....

Results

Characteristics of a cohort

2. You included DSD enrolment numbers. Being an important LTFU intervention, it is not clear why this variable was not included in the LTFU risk model.

3. Up to 47% of patients were categorised in WHO Stage III and IV. This is a much higher proportion than would be expected from a mature HIV program. What is the reason for this, especially since these would be likely categorised as Advanced HIV Disease with enhanced interventions including those targeting LTFU.

4. Age categories - The age categories analysed stop at 45yrs and above. It is not clear why older age categories were grouped together, as opposed to maintaining the age bands or at least to 50yrs and above. See some examples in published literature (https://doi.org/10.1186/s12879-022-07505-0;
https://doi.org/10.1186/s12889-020-8426-1).

5. Correct the spelling of under-nourished in Table 4.

6. The protective effect of WHO Stage III/IV on LTFU is an outlier. Would this be the same if you looked at each WHO Stage as a variable? See the the shared examples above.

Discussion

7. Line 266-269 and line 283-285. Results are repeated in the discussion section. I suggest that since these are already mentioned elsewhere, to only limit to what would be key for comparing and contrasting with published literature.

Reviewer #2: Given that LTFU poses a serious threat to treatment outcomes, particularly viral suppression and the prevention of disease progression, this work is relevant. I commend the authors for their efforts and would like to offer the following suggestions:

1. The definition of LTFU used in the manuscript appears to omit a key conceptual component. LTFU, by standard practice, excludes individuals with documented outcomes such as confirmed death, formal transfer to another facility, or medical discontinuation of ART. These are not considered competing events but rather exclusions from the LTFU classification itself.

It is important to check the records of patients labeled as LTFU. Some may have died or transferred without documentation. Identifying these cases helps to avoid misclassification and improves competing risk analysis. I suggest that the authors clarify this distinction and align their operational definition accordingly.

2. The current operational definition of LTFU in the study could lead to an overestimation of its incidence. In many HIV treatment programs, patients are dispensed ART for multiple months (e.g., 90 days) and are not necessarily required to return for a clinic visit during this period unless clinically indicated. Though some clinicians might schedule interim assessments for monitoring purposes. If such a patient misses an interim appointment but completes their medication course within the dispensation period, classifying them as LTFU based solely on missed interim visits may result in misclassification. For this reason, most studies define LTFU in reference to the next scheduled medication refill, not just any missed appointment. I suggest that the authors revisit their LTFU definition and consider this operational nuance to improve accuracy.

3. It would be helpful if the authors could specify the nature of the cohort used in the study. Was this an open cohort (allowing for rolling enrollment) or a closed cohort (fixed enrollment period)? Clarifying this will enhance the transparency of the methodology and allow for better interpretation of follow-up time and incidence rates.

4. For improved clarity and to aid in interpretation, I recommend that the authors present the results of the univariate analysis (CHR and SHR) prior to the adjusted multivariate models. This will allow readers to better understand the effect of covariates before and after adjustment.

**Do you want your identity to be public for this peer review?** For information about this choice, including consent withdrawal, please see our Privacy Policy

Reviewer #1: **Yes: ** Paul Wekesa

Reviewer #2: No

---

## [Author Response · Author response to Decision Letter 1]

7 Jul 2025

Response to Reviewers

Subject: Revised Manuscript Submission

PONE-D-25-16597

Time-dependent predictors of loss to follow-up in HIV care in a low-resource settings: A competing risks approach

Dear Dr. Hamufare Mugauri,

Thank you for the opportunity to revise our manuscript. We sincerely appreciate the time and thoughtful feedback provided by you (editors) and the reviewers.

We have carefully addressed all comments raised by the academic editor and reviewers, and we are submitting the following revised materials for your consideration:

• A detailed rebuttal letter addressing each point raised by the editor and reviewers. This document is uploaded as a separate file labeled “Response to Reviewers.”

• A marked-up version of the manuscript showing all changes made from the original submission. This file is labeled “Revised Manuscript with Track Changes.”

• A clean version of the revised manuscript without tracked changes, labeled “Manuscript.”

• Updated supporting information.

We hope the revised manuscript meets your expectations and we look forward to your further evaluation and feedback.

Sincerely,

Tamrat Endebu, PhD Candidate

Point raised by the academic editor

Response: Thank you for your guidance. We have made efforts to ensure compliance with PLOS ONE’s style requirements, including the file naming conventions.

Response: Thank you for your valuable comment. We have now provided additional details as “All data were fully anonymized prior to access, and the requirement for informed consent was waived by the ethics committee, as the study involved retrospective analysis of deidentified data. We strictly maintained confidentiality by using anonymized records solely for research purposes.” (Manuscript: Line 161-163).

3. We note that your Data Availability Statement is currently as follows: [All relevant data are within the manuscript and its supporting information files.]

Response: Thank you for your guidance. We confirm that we have included the minimal data set required to replicate the study findings. The de-identified dataset used for analysis has been uploaded as a Supporting Information file (S3 Data.dta). These materials provide all values used to compute the reported summary statistics, regression outputs, and figures.

Reviewers' comments:

Reviewer's Responses to Questions

Comments to the Author

1. Is the manuscript technically sound, and do the data support the conclusions?

Reviewer #1: Yes

Reviewer #2: Yes

2. Has the statistical analysis been performed appropriately and rigorously?

Reviewer #1: Yes

Reviewer #2: Yes

3. Have the authors made all data underlying the findings in their manuscript fully available?

Reviewer #1: Yes

Reviewer #2: Yes

4. Is the manuscript presented in an intelligible fashion and written in standard English?

Reviewer #1: Yes

Reviewer #2: Yes

5. Review Comments to the Author

Point raised by the reviewers

Reviewer #1: I have enjoyed reading your well written manuscript on time-dependent predictors of LTFU in Ethiopia. Please see a few comments for your consideration.

Data processing and analysis

1. Line 139 - Consider using plural for data. Data were....

Response: Thank you for your overall positive feedback and valuable comments. We have edited as “Data were” (Manuscript: Line 133).

Results

Characteristics of a cohort

2. You included DSD enrolment numbers. Being an important LTFU intervention, it is not clear why this variable was not included in the LTFU risk model.

Response: Thank you for your valuable observation. We agree that DSD models are central to HIV care strategies and are often designed to reduce LTFU. While DSD was originally introduced as an intervention to improve retention, emerging evidence suggests that the type of DSD model itself may influence patient outcomes differently. Some studies have reported that certain DSD models, particularly those with shorter refill intervals (e.g., 3MMD), may not offer the same retention benefits as longer-term models like ASM/6MMD. Moreover, patients not enrolled in any DSD model may face structural or clinical barriers that place them at higher risk of disengagement from care. Given this, we considered DSD enrollment status a factor of LTFU and incorporated it into our risk models.

3. Up to 47% of patients were categorised in WHO Stage III and IV. This is a much higher proportion than would be expected from a mature HIV program. What is the reason for this, especially since these would be likely categorised as Advanced HIV Disease with enhanced interventions including those targeting LTFU.

Response: Thank you for your insightful comment. We acknowledge that the proportion of patients categorized as WHO Stage III/IV appears higher than expected in the context of a mature HIV program. However, several factors may explain this observation in our study. First, the dataset includes patients who initiated ART over the past five years, and for many, clinical staging was assessed at or near ART initiation, a period during which individuals commonly present with more advanced disease. This is particularly true in low-resource settings where delayed diagnosis and late initiation of ART remain challenges. Second, for patients retained in care, WHO clinical stage was recorded at the time of last contact. In contrast, for those who were LTFU (27%), died (8.4%), or transferred out (12.9%), WHO staging data were primarily recorded at the time of last contact or enrollment, with only a few having updates at their last contact. We have clarified this in the Methods section of the manuscript (Manuscript: Line 110-112). Given that LTFU, transfer out, and death often occur during the early months of ART, when many patients still present with advanced disease, this likely contributed to the higher proportion of Stage III/IV classifications observed in our cohort.

4. Age categories - The age categories analysed stop at 45yrs and above. It is not clear why older age categories were grouped together, as opposed to maintaining the age bands or at least to 50yrs and above. See some examples in published literature (https://doi.org/10.1186/s12879-022-07505-0;
https://doi.org/10.1186/s12889-020-8426-1).

Response: Thank you for your insightful comment regarding the age categorization. We acknowledge that more granular age groupings, such as 45–49 and ≥50 years, are frequently used in the literature and can provide additional nuance. In our analysis, we categorized age into 15–24, 25–34, 35–44, and ≥45 years to maintain adequate statistical power, particularly for stratified analyses, where further disaggregation would have resulted in small cell sizes and potentially unstable estimates. We agree that future studies with larger sample sizes would benefit from employing more detailed age stratifications to enhance interpretability and comparability.

5. Correct the spelling of under-nourished in Table 4.

Response: Thank you for your observation. We have corrected the spelling to “undernourished” in Table 4.

6. The protective effect of WHO Stage III/IV on LTFU is an outlier. Would this be the same if you looked at each WHO Stage as a variable? See the shared examples above.

Response: Thank you for your valuable comment and thoughtful suggestion. We had already noted and discussed the unexpected protective association between advanced WHO clinical stages (III/IV) and LTFU in our initial analysis. As stated in the discussion section, our findings revealed that patients in WHO Stage III/IV had a significantly lower risk of LTFU compared to those in Stages I/II. This finding aligns with some previous studies while also contradicts with other several studies (Manuscript: Line 352-362). Based on your helpful suggestion, we re-analyzed the data using each WHO stage as a separate variable. However, the pattern remained largely consistent, with lower LTFU risk among patients in advanced stages. This finding may reflect context-specific programmatic factors and warrants further investigation.

Discussion

7. Line 266-269 and line 283-285. Results are repeated in the discussion section. I suggest that since these are already mentioned elsewhere, to only limit to what would be key for comparing and contrasting with published literature.

Response: Thank you very much for your insightful observation and helpful suggestion. We have revised the discussion section by removing the repeated results and focused instead on comparing our findings with existing literature (Manuscript: Lines 263–265 and 279–280).

Reviewer #2

Reviewer #2: Given that LTFU poses a serious threat to treatment outcomes, particularly viral suppression and the prevention of disease progression, this work is relevant. I commend the authors for their efforts and would like to offer the following suggestions:

1. The definition of LTFU used in the manuscript appears to omit a key conceptual component. LTFU, by standard practice, excludes individuals with documented outcomes such as confirmed death, formal transfer to another facility, or medical discontinuation of ART. These are not considered competing events but rather exclusions from the LTFU classification itself.

It is important to check the records of patients labeled as LTFU. Some may have died or transferred without documentation. Identifying these cases helps to avoid misclassification and improves competing risk analysis. I suggest that the authors clarify this distinction and align their operational definition accordingly.

Response: Thank you for your valuable comment and for highlighting this important conceptual distinction. We fully agree that, by standard practice, LTFU excludes individuals with documented outcomes such as confirmed death, formal transfer out, or medical discontinuation of ART. In our analysis, we adhered to this convention: only patients without any formal documentation of transfer, death, or treatment discontinuation, and who had interrupted their scheduled visit by at least 28 days, were classified as LTFU. As the reviewer correctly points out, some individuals recorded as LTFU may, in fact, have experienced undocumented deaths or unrecorded transfers. We have now clarified the operational definition of LTFU in the Methods section (Lines 90–93). In addition, while our use of a competing risks framework helps to mitigate potential bias from misclassification by modeling death and transfer as competing events, we acknowledge that imperfect data capture remains a limitation in retrospective routine datasets (Manuscript: Lines 364–367).

2. The current operational definition of LTFU in the study could lead to an overestimation of its incidence. In many HIV treatment programs, patients are dispensed ART for multiple months (e.g., 90 days) and are not necessarily required to return for a clinic visit during this period unless clinically indicated. Though some clinicians might schedule interim assessments for monitoring purposes. If such a patient misses an interim appointment but completes their medication course within the dispensation period, classifying them as LTFU based solely on missed interim visits may result in misclassification. For this reason, most studies define LTFU in reference to the next scheduled medication refill, not just any missed appointment. I suggest that the authors revisit their LTFU definition and consider this operational nuance to improve accuracy.

Response: Thank you for your thoughtful comment regarding the operational definition of LTFU. We agree that defining LTFU solely based on missed clinic visits without accounting for multi-mont

---

## [Editor Report · Decision Letter 1]

Time-dependent predictors of loss to follow-up in HIV care in a low-resource settings: A competing risks approach

PONE-D-25-16597R1

Dear Dr. Gebre,

We’re pleased to inform you that your manuscript has been judged scientifically suitable for publication and will be formally accepted for publication once it meets all outstanding technical requirements.

Kind regards,

Hamufare Dumisani Mugauri, Ph.D. Medicine and Health Sciences

Academic Editor

PLOS ONE

---

## [Editor Report · Acceptance letter]

PONE-D-25-16597R1

PLOS ONE

Dear Dr. Gebre,

I'm pleased to inform you that your manuscript has been deemed suitable for publication in PLOS ONE. Congratulations! Your manuscript is now being handed over to our production team.

Kind regards,

on behalf of

Dr Hamufare Dumisani Mugauri

Academic Editor

PLOS ONE